ROCK activity regulates functional tight junction assembly during blastocyst formation in porcine parthenogenetic embryos

Kwon Jeongwoo 1
Kim Nam-Hyung 1 nhkim@chungbuk.ac.kr
Choi Inchul 2 icchoi@cnu.ac.kr
1 Department of Animal Science, Chungbuk National University , Cheongju Chungbuk , South Korea
2 Department of Animal and Dairy Sciences, Chungnam National University , Daejeon Chungnam , South Korea
Sun Shao-Chen
Electronic publication date: 2016 Apr 11
Publication date: 2016
Volume: 4
Electronic Location ID: e1914
Received 2016 Feb 12; Accepted 2016 Mar 18
Copyright: ©2016 Kwon et al.
Copyright year: 2016
Copyright holder: Kwon et al.
License: This is an open access article distributed under the terms of the Creative Commons Attribution License, which permits unrestricted use, distribution, reproduction and adaptation in any medium and for any purpose provided that it is properly attributed. For attribution, the original author(s), title, publication source (PeerJ) and either DOI or URL of the article must be cited.
License URL: https://creativecommons.org/licenses/by/4.0/

Keywords: Preimplantation, Porcine embryo, Blastocyst, Tight junction

Funding: Next-Generation BioGreen21 Program PJ011213 PJ011126 Rural Development Administration (RDA), Republic of Korea This was supported by the Next-Generation BioGreen21 Program (CABX, Project No. PJ011213 to IC, and PJ011126 to NK), the Rural Development Administration (RDA), Republic of Korea. The funders had no role in study design, data collection and analysis, decision to publish, or preparation of the manuscript.

==============================
The Rho-associated coiled-coil-containing protein serine/threonine kinases 1 and 2 (ROCK1 and ROCK2) are Rho subfamily GTPase downstream effectors that regulate cell migration, intercellular adhesion, cell polarity, and cell proliferation by stimulating actin cytoskeleton reorganization. Inhibition of ROCK proteins affects specification of the trophectoderm (TE) and inner cell mass (ICM) lineages, compaction, and blastocyst cavitation. However, the molecules involved in blastocyst formation are not known. Here, we examined developmental competence and levels of adherens/tight junction (AJ/TJ) constituent proteins, such as CXADR, OCLN, TJP1, and CDH1, as well as expression of their respective mRNAs, after treating porcine parthenogenetic four-cell embryos with Y-27632, a specific inhibitor of ROCK, at concentrations of 0, 10, 20, 100 µM for 24 h. Following this treatment, the blastocyst development rates were 39.1, 20.7, 10.0, and 0% respectively. In embryos treated with 20 µM treatment, expression levels of CXADR, OCLN, TJP1, and CDH1 mRNA and protein molecules were significantly reduced (P < 0.05). FITC-dextran uptake assay revealed that the treatment caused an increase in TE TJ permeability. Interestingly, the majority of the four-cell and morula embryos treated with 20 µM Y-27643 for 24 h showed defective compaction and cavitation. Taken together, our results indicate that ROCK activity may differentially affect assembly of AJ/TJs as well as regulate expression of genes encoding junctional proteins.

Introduction

The Rho-associated coiled-coil-containing protein serine/threonine kinases (ROCK, referring collectively to both ROCK1 and ROCK2 isoforms) are downstream effectors of the Rho subfamily GTPase. They regulate cell migration, intercellular adhesion, cell polarity cytokinesis, and cell proliferation by affecting actin cytoskeleton reorganization and microtubule dynamic (Etienne-Manneville & Hall, 2002; Olson & Sahai, 2008; Ridley, 1999). The biological roles and expression patterns of ROCK in mammalian embryo development have been intensively studied by pharmacological disruption of ROCK activity by the specific ROCK inhibitor Y-27632 and by using Real-Time qRT-PCR (Quantitative Reverse Transcription Polymerase Chain Reaction) and immunocytochemistry (ICC) approaches to visualize expression of ROCK genes and proteins (Duan et al., 2014; Kawagishi et al., 2004; Zhang et al., 2014). These studies demonstrated that early embryo developmental competence was affected in a dose/stage-dependent manner when mouse and porcine embryos were exposed to Y-27632 (100 µM in mouse, 5, 10, and 15 µM in porcine). For example, both ROCK1 and ROCK2 were detected during porcine preimplantation embryo development, but ROCK2 was more abundantly expressed in the trophectoderm (TE) of the blastocyst embryos compared to the expression levels of ROCK1. The majority of early cleavage stage embryos failed to progress to the blastocyst stage and remained arrested at the compaction/morula stages after they were exposed to Y-27632 at concentrations higher than 10 µM. Furthermore, porcine and mouse morula embryos treated with progressively higher concentrations of Y-27632 did not develop to the blastocyst stage either.

Recently, it has been reported that pharmacological inhibition of ROCK and RHO by Y-27632 and RHO inhibitor I from the two-cell stage enhanced the inner cell mass (ICM) lineage and suppressed TE lineage establishment through activation of Hippo signaling and disruption of cellular polarity due to suppression of the activity of the LATS kinase in the outer blastomere cells (Kono, Tamashiro & Alarcon, 2014). Furthermore, treatment of blastocysts with Y-27632 disrupted ICM morphology and caused a transient reduction in the blastocyst cavity size and fetal loss after implantation (Laeno, Tamashiro & Alarcon, 2013). In line with these findings, inhibition of ROCK activity by Y-27632 induced defects of position-dependent HIPPO signaling and downregulation of Cdx2 expression in the emerging TE lineage (Cao et al., 2015).

Taken together, these previous studies provided strong evidence that ROCK activity is involved in blastocyst formation by regulating TE and ICM establishment via position-dependent HIPPO signaling. However, it is not known why formation of the fluid-filled cavity is inhibited in early cleaving embryos exposed to the ROCK inhibitor and why the size of the blastocyst cavity transiently decreased in blastocysts treated with Y-27632. It is well documented that blastocyst formation requires water channels as physiological mediators of fluid movement across the TE, a correct Na+∕K+− ATPase-generated trans-TE ion gradient for water accumulation, and a proper assembly of tight junction (TJ) proteins (Watson & Barcroft, 2001; Watson, Natale & Barcroft, 2004). Recently, ROCK has been reported to be down-regulated in TFAP2C-depleted embryos that were arrested in the transition between morula and blastocyst stages. These embryos exhibited defects of paracellular sealing because of the TJ disruption and had a phenotype resembling that of embryos treated with Y-27632 (Cao et al., 2015; Choi et al., 2012). Thus, we hypothesized that inhibition of ROCK activity leads to the impairment of TJ functions. Particularly, our recent finding that CXADR is required for adherens junction (AJ) and TJ assembly during porcine blastocyst formation (Kwon, Kim & Choi, 2016) also supports this hypothesis, because CXADR was reported to be directly associated with ROCK in human tumor cells (Saito et al., 2014). Here, we report that ROCK activity is involved in functional TJ assembly and paracellular sealing during the preimplantation development of parthenogenetic porcine embryos.

Materials and Methods

All reagents were purchased from Sigma-Aldrich (St. Louis, MO, USA) unless stated otherwise.

Collection of porcine oocytes and embryo culture

Porcine-oocyte collection and embryo culture were carried out as described previously (Lee et al., 2015). Briefly, ovaries obtained from a local slaughter house were transported to the laboratory in Dulbecco’s phosphate-buffered saline (DPBS) at 37 °C. Cumulus-oocyte complexes (COCs) aspirated from 3–6 mm follicles were washed three times with 4-(2-hydroxyethyl)-1-piperazineethanesulfonic acid (HEPES)-buffered Tyrode’s medium containing 0.1% (w/v) polyvinyl alcohol (HEPES-TL-PVA). Groups of ∼50 COCs were incubated in the in vitro maturation medium for 44 h at 39 °C and then denuded by pipetting in HEPES-TL containing 1 mg/mL hyaluronidase for 2–3 min. Denuded oocytes were treated with 50 µM calcium ionophore A23187 for 5 min and incubated in porcine zygote medium 3 (PZM3) containing 7.5 mg/mL cytochalasin B for 3 h for parthenogenetic activation. The embryos were washed three times in HEPES-TL-PVA, transferred to PZM3 supplemented with 0.4% (w/v) BSA, and cultured until use at 39 °C in a humidified atmosphere containing 5% of CO2.

Quantification of transcript levels

Isolation of mRNA from porcine blastocysts (ten samplings per each biological replicate) was carried out by using the Dynabeads mRNA Direct Kit (Dynal ASA, Oslo, Norway). First-strand cDNA was synthesized with the Superscript Reverse Transcriptase Enzyme (Invitrogen, Grand Island, NY, USA). Quantitative real-time RT-PCR (qRT-PCR) was carried out using cDNA synthesized with the Superscript Reverse Transcriptase Enzyme (Invitrogen, Grand Island, NY, USA) on a DNA Engine Opticon 2 Fluorescence Detection System (MJ Research, Waltham, MA, USA) with the DyNAmo SYBR Green qPCR Kit (Finnzymes Oy, Espoo, Finland). Transcripts of porcine CXADR, OCLN, and TJP1 genes were amplified using specific primer pairs and conditions (Kwon, Kim & Choi, 2016). Relative quantification of gene expression was performed by the 2−ΔΔCt method from three technical and biological replicates for the control and ROCK inhibitor-treated groups. GAPDH was used as the internal control in all experiments.

Immunocytochemistry (ICC)

The procedure was carried out as described previously (Lee et al., 2015). Porcine preimplantation embryos were washed in DPBS that contained polyvinyl alcohol (1 mg/mL) and fixed for 20 min in 3.7% (w/v) paraformaldehyde dissolved in DPBS. Thereafter, embryos were permeabilized and blocked in DPBS that contained 0.5% (v/v) Triton X-100 and 5% donkey serum at room temperature for 1 h. Embryos were then incubated with a rabbit polyclonal anti-CXADR antibody (Sigma), a mouse monoclonal anti-TJP1 antibody (ZO-1; Zymed, San Francisco, CA, USA), or a mouse monoclonal anti-OCLN antibody (Zymed) in the blocking solution (DPBS with 5% donkey serum) overnight at 4 °C, followed by incubation with Alexa Fluor 488- and 594-conjugated antibodies (Molecular Probes, Eugene, OR, USA) as secondary antibodies. The embryos were then mounted in Vectashield containing 4′,6-diamidino-2-phenylindole (DAPI; Vector Laboratories, Burlingame, CA, USA) and analyzed by a laser-scanning confocal system with a krypton-argon ion laser under a Leica DM IRB inverted microscope (DM IRB; Leica, Wetzlar, Germany).

Treatment of embryos with the ROCK inhibitor Y-27632

To examine the effects of the ROCK inhibitor Y-27632 on embryo development and polarized distribution of AJ and TJ proteins, late four-cell and morula embryos were incubated in 30 µl droplets supplemented with Y-27632 at final concentrations of 10, 20, and 100 µM for 24 h at 39 °C in a humidified atmosphere containing 5% of CO2. Since the stock solution of Y-27632 was prepared in dimethyl sulfoxide (DMSO), control embryos were treated with the equivalent amount of DMSO and cultured until use.

TJ permeability assay using FITC-dextran uptake

To assess the effects of inhibition of ROCK activity on TJ permeability, DMSO vehicle control embryos and blastocysts derived from morula embryos treated with 10 µM and 20 µM of Y-27632 were incubated in the culture medium containing 1 mg/mL 40-kDa (fluorescein isothiocyanate) FITC-dextran for 30 min at 37 °C. Then, they were immediately washed several times in 50-µl droplets of PZM5 to remove FITC-dextran on the surface of embryos, placed into a clean droplet of PZM5, and visualized under an inverted fluorescence microscope (TE2000U; Nikon, Tokyo, Japan).

Statistical analysis

Quantitative development data as well as results of protein and gene expression experiments were subjected to the analysis of variance (ANOVA), whereas data from the tight junction permeability assay were analyzed by the chi-square test using the Statistical Analysis System software (Statistical Analysis System, Inc., Cary, NC, USA). The data are presented as the mean ± standard error of the mean (SEM). Differences with P values <0.05 were considered statistically significant unless stated otherwise.

Results

Blastocyst development of porcine embryos treated with the specific ROCK inhibitor Y-27632

To evaluate the effects of ROCK inhibition on blastocyst development, we treated the four-cell embryos with 0 (control), 10, 20, and 100 µM Y-27632 for 24 h. We observed that exposure to the inhibitor led to significantly lower blastocyst development rates (39.1, 20.7, 10.0, and 0%, respectively) (Fig. 1A). The effects were significantly greater at higher Y-27632 concentrations (20 and 100 µM). The embryos that failed to develop to the blastocyst stage were arrested at the pre-compaction stage in presence of 10 and 20 µM Y-27632 (Figs. 1B and 2A).

Figure 1 Blastocyst development of the four-cell porcine embryos treated with ROCK inhibitor, Y-27632.

(A) The blastocyst development rates of embryos exposed to different concentrations of Y-27632 (control 0, treatment 10, 20, and 100 µM) for 24 h at the four-cell stage (77, 70, 74, and 70 embryos, respectively; 22–27 embryos/ replicate). Error bar means ± SEM, different letters are significantly different (P < 0.05). n.d. not detected. (B) Representative images of embryos treated with the different concentrations at 48 h.p.a (hours post activation) and the embryos cultured until 144 h.p.a (at expanding stages) after 24 h treatment of the four cell embryos with different concentrations of Y-27632. Bar 150 µM.

Figure 2 Stage specific effects of ROCK inhibitor on compaction and cavitation.

(A) The percentages of 92 and 71 embryos treated with 20 µM Y-27632 at the four-cell or morula stages, respectively that developed to morula, and blastocyst at 96 and 120 h.p.a,. (B) Representative images of porcine embryos at 120 h.p.a. following treatment of the four-cell and morula, respectively, with 20 µM Y-27632 for 24 (C) Total cell number of embryos at 120 h.p.a after treatment of morula embryos with 20 µM Y-27632; For this analysis, blastomeres of 24 morula embryos were counted in the control and treatment group. Error bar means ± SEM, Astrisks(*) indicate means that are significantly different from the control (P < 0.05). (D) Localization of CXADR protein in porcine embryos. CXADR proteins were localized to cell to cell boundaries in the control compacted embryo and the blastocyst but in 20 µM Y-27632 treated embryos was disused to cytoplasm or not distinct continuous lines on apical regions of each blastomeres.

Next, we treated the four-cell and morula embryos with 20 µM Y-27632 for 24 h and examined the stage-dependent effects of the ROCK inhibitor on morphological responses such as compaction and cavitation at 120 h.p.a (hours post activation). The majority of the four-cell embryos did not develop beyond the compacted/morula stage and remained at the four-cell stage as a result of treatment (Figs. 2A and 2B). Furthermore, while 49.1% of control morula reached the blastocyst stage with cavity and expansion, only 9.2% of morula treated with the ROCK inhibitor developed to the blastocyst (Fig. 2B). At 120 h.p.a, we found that the total cell number in embryos treated with 20 µM Y-27632 was lower than that in control embryos (16.25 ± 2.1 and 29.7 ± 4.4, respectively) (Fig. 2C).

We also investigated the subcellular localization of the CXADR protein, which has been reported to be associated with ROCK activity, to determine whether ROCK affects polarized distribution of adhesion and tight junction proteins. In embryos arrested at the precompaction stage by their exposure to 100 µM Y-27632 for 24 h, the fluorescent signals were dispersed in the cytoplasm and very weak signals were detected at 48 h after the treatment, but control embryos developed to the morula stage and showed continuous thin lines at the apical region of cells (Fig. 2D). Embryos treated with the inhibitor at the morula stage for 24 h showed relatively visible continuous lines along the cell–cell boundaries compared to those in the four-cell treatment group but less distinct than those in control blastocysts at 120 h.p.a (Fig. 2D). These results suggested that ROCK activity affects cellular polarization of adhesion and tight junction proteins.

Inhibition of AJ and TJ assembly in embryos by Y-27632

Based on our findings and previous reports regarding the role of ROCK in blastocyst formation (Choi et al., 2012; Kawagishi et al., 2004; Zhang et al., 2014), we speculated that inhibition of the cavitation onset in Y-27632-treated embryos is caused by mislocalization or AJ and TJ proteins and impaired TJ assembly. Thus, we examined localization and expression levels of CXADR, OCLN, TJP1, and CDH1 mRNA and protein molecules, which are essential for the integrity of the AJ-TJ complex, by using ICC and qRT-PCR at the blastocyst stage. In embryos exposed to 20 µM Y-27632, we observed that the CXADR and TJP1 proteins were barely detected at the apical regions of the cell–cell boundaries. In addition, while the OCLN protein was expressed in continuous lines at the apical edge of cell–cell contacts, its total expression in the treated blastocyst was very weak (Fig. 3A). The relative levels of CXADR, OCLN, TJP1, and CDH1 mRNA transcripts were also significantly lower in embryos from the Y-27632 treatment group (P < 0.05, Fig. 3B).

Figure 3 Expression of tight junction and adherens proteins.

(A) Localization of tight junction associated proteins (CXADR, OCLN, and TJP) and adherens protein (CDH1) in control and 20 µM Y-27632 treated embryos. All tight junction associated proteins and CDH1 were reduced in the treatment group. (B) Relative transcription levels of CXADR, OCLN TJP1, and CDH1 of embryos treated with 10 µM Y-27632 to non-treated control blastocyst (ten per biological and three times technical replicates). Error bar means ± SEM, Astrisks(*) indicate means that are significantly different from the control (P < 0.05). RQ (Relative Quantification).

These findings led us to hypothesize that integrity of the TJ complex was disrupted by mislocalization and lower expression levels of tight junction proteins in presence of the ROCK inhibitor. To examine the functional barrier properties of the tight junction assembly, we treated morula embryos with 10 µM and 20 µM Y-27632 for 24 h and cultured them until the blastocyst stage. At 144 h.p.a, we exposed blastocysts, including control ones treated with DMSO, to 40-kDa FITC-dextran. We found a significant difference in the tight junction permeability in blastocysts from control and inhibitor-treated groups (P < 0.05), but no difference between the two treatment groups (Table 1). This observation demonstrated that inhibition of ROCK activity led to impairment of tight junction assembly during porcine blastocyst cavitation.

Table 1 FITC uptake assay after treatment of Y-27632 at the morula stage.

FITC-Dextrans uptake assay for paracellular sealing in blastocyst. Y-27632 treated embryos showed that defects of paracellular sealing compared with control embryos.

	DMSO	10 μM	20 μM	
FITC dextran Positive	3	7	10	
FITC dextran Negative	59	25	21	
Notes.

Y-27632 treatment at the morula stage

Discussion

Previous studies reported that inhibition of ROCK activity by its specific inhibitor Y-27632 prevented blastocyst cavitation, tight junction biogenesis, TE and ICM formation, establishment of the intracellular polarity in the form of the apical-basal axis, and affected developmental competency in a dose- or a stage-dependent manner in mouse (Cao et al., 2015; Duan et al., 2014; Kawagishi et al., 2004; Kono, Tamashiro & Alarcon, 2014). The defects found in embryos treated with the ROCK inhibitor were attributed to the loss of position-dependent HIPPO signaling that is mediated via the RHO-ROCK signaling (Kono, Tamashiro & Alarcon, 2014). A number of studies have shown that ion gradient, adherens junctions, tight junctions, and water channels are properly expressed and localized on the outer cells of embryos and the TE to form blastocyst (reviewed in Choi et al., 2012; Watson & Barcroft, 2001; Watson, Natale & Barcroft, 2004). However, the specific cellular events and molecules involved in developmental perturbations in early embryos induced by ROCK inhibition were not defined. Here, focusing on tight junction assembly, we investigated how inhibition of ROCK activity affected preimplantation development of porcine embryos.

First, we demonstrated that treatment with Y-27632 resulted in the failure of early porcine embryo development in a dosage-dependent manner. In line with previous results obtained in the mouse model (Duan et al., 2014), treatment with Y-27632 (100 µM) prevented four-cell embryos from undergoing compaction and also inhibited the morula-to-blastocyst transformation during porcine preimplantation development (summarized in Fig. 4A). A possible explanation of defective compaction and impaired blastocyst formation is that the two ROCK isoforms, ROCK1 and ROCK2, have different stage-specific expression patterns and subcellular localizations during preimplantation development and, therefore, can have different biological activities (Zhang et al., 2014). ROCK1 is predominantly detected in the cytoplasm and localized mainly at the cell–cell boundaries at the four-cell to morula stages. However, ROCK1 is barely detected in the blastocyst, while ROCK 2 is present in both the cytoplasm and the nucleus with predominant expression in the TE nuclei (Zhang et al., 2014) (see schematic representation of expression and localization of ROCK1 and ROCK2 in porcine embryos in Fig. 4). Considering the spatial and temporal expression patterns of ROCK1 and ROCK2 and the stage-dependent impairment of embryo development by ROCK inhibition, we suggest that the arrest of embryos at pre-compaction is associated with inhibition of ROCK1 rather than ROCK2, because ROCK1 is involved in cell adhesion via regulating actomyosin contraction and association with E-cadherin complexes (Shi et al., 2013; Smith et al., 2012).

Figure 4 Summary of effects of ROCK inhibition on embryo development and model of regulation of ROCK activity in porcine embryos.

(A) Schematic view of embryo development and the results of ROCK inhibition experiments. Yellow and red (cell polarity); hatched bar (DMSO treatment); filled bar (Y-27632 treatment). Green (expression of ROCK1 or ROCK2); Circle (white: cell membrane; center: nucleus; others: cytoplasm), adapted from Zhang et al., 2014. (B) Model of ROCK activitiy in porcine embryos. Cytoplasmic ROCK1 is involved in phosphorylation of OCLN and linked to actin stability, providing establishment of tight junction, and paracellular sealing during morula to blastocyst transition. Tight junction also affects expression of genes related with tight junction including OCLN, TJP1, and CXADR via nuclear ROCK2 mediated by p300, histone acetyltransferase at the blastocyst to maintain junctional complexes.

Next, we examined subcellular localization of CXADR, which was recently reported to be required for adherens and tight junction formation during porcine blastocyst development (Kwon, Kim & Choi, 2016) and shown to directly associated with ROCK1 and ROCK2 in human carcinoma cells (Saito et al., 2014). The observed disruption of localization of the CXADR protein and reduced expression of the CXADR gene indicated that clustering of junctional proteins complex and its functional barrier can be mediated by CXAD.R. Thus, we examined single-pass membrane protein (CXADR), four-pass membrane domain (OCLN), and cytoplasmic plaque protein (TJP1), and adherens protein (CDH1), found a significant decrease in the protein and mRNA expression levels, and demonstrated increased paracelluar permeability in the ROCK inhibitor treated blastocysts (Table 1). The deficient barrier function in blastocysts was likely caused by the disruption of tight junction assembly and deregulation of gene expression on the morula-to-blastocyst developmental stage. A growing body of evidence has demonstrated that adherens and tight junctions are both linked intracellularly to F-actin via adaptor proteins such as TJP1 and OCLN, which are phosphorylated by ROCK (Aijaz, Balda & Matter, 2006; Balda et al., 1996; Gopalakrishnan et al., 1998; Hartsock & Nelson, 2008; Hirase et al., 2001; Ohnishi et al., 2004; Sahai & Marshall, 2002; Wittchen, Haskins & Stevenson, 1999; Yamamoto et al., 2008). In addition, nuclear ROCK2 has been reported to be involved in transcriptional regulation of the trophoblast lineage and co-localized with the p300 acetyltransferase (Carey et al., 2013; Rodriguez et al., 2004; Tanaka et al., 2006). Converging lines of evidence suggest that inhibition of cytoplasmic ROCK1 activity in morula can adversely affect the interaction of cytoskeleton F-actin with TJP1 and phosphorylation of the carboxy-terminal domain of OCLN, which is important for paracellular sealing of the TE during the transition stage (Fig. 4B). Furthermore, inhibition of nuclear ROCK2 activity may lead to reduced expression of the CXADR, OCLN, TJP, and CDH1 genes via transcriptional regulation associated with p300 activity (Fig. 4B).

In summary, we demonstrated that inhibition of ROCK activity impairs early porcine embryo developmental steps such as compaction and cavitation. We suggest that observed defective embryo development and lower gene expression of AJ/TJ associated genes in presence of the specific ROCK inhibitor are caused by specific temporo-spatial inhibition of distinct ROCK1 and ROCK2 functions in porcine embryos. In this context, it will be important to establish the molecular mechanisms of differential subcellular targeting of ROCK isoforms during intracellular polarization and adherens/tight junction biogenesis throughout early embryo development.

Supplemental Information

Supplemental Information 1 Fig1A raw data

The blastocyst development rates of the four-cell stage embryos exposed to different concentrations of Y-27632 (control 0, treatment 10, 20, and 100 µM).

Click here for additional data file.

Supplemental Information 2 Fig2A raw data

The fractions of embryos treated with 20 µM Y-27632 at the four-cell or morula stages that developed to the morula and blastocyst stages, respectively, at 96 and 120 h.p.a.

Click here for additional data file.

Supplemental Information 3 Fig2C raw data

Total cell number of embryos at 120 h.p.a after the treatment of morula embryos with 20 µM Y-27632.

Click here for additional data file.

Supplemental Information 4 Fig3b raw data

Relative levels of CXADR, OCLN, TJP1, and CDH1 transcripts in embryos treated with 10 µM Y-27632 in comparison to their levels in non-treated, control blastocysts.

Click here for additional data file.

Additional Information and Declarations

Competing Interests

Author Contributions

Data Availability

The authors declare there are no competing interests.

Jeongwoo Kwon conceived and designed the experiments, performed the experiments, analyzed the data, wrote the paper, prepared figures and/or tables, reviewed drafts of the paper.

Nam-Hyung Kim conceived and designed the experiments, contributed reagents/materials/analysis tools, wrote the paper, reviewed drafts of the paper.

Inchul Choi conceived and designed the experiments, analyzed the data, contributed reagents/materials/analysis tools, wrote the paper, prepared figures and/or tables, reviewed drafts of the paper.

The following information was supplied regarding data availability:

The raw data has been provided as Supplemental Information.

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
