# Peer review of "ROCK activity regulates functional tight junction assembly during blastocyst formation in porcine parthenogenetic embryos"

_PeerJ, doi:10.7717/peerj.1914_

## Round 0.1 · original submission · Minor Revisions

Please revise the manuscript following the comments of the reviewers.

Reviewer 1 ·

Basic reporting

No comments

Experimental design

No comments

Validity of the findings

No comments

Additional comments

In this study the authors showed that ROCK activity is required for early embryonic development of porcine parthenotes. Moreover, they showed that inhibition of ROCK impairs expression of adherens/tight junction constituent proteins, including CXADR, OCLN, TJP1 and CDH1. Overall this manuscript is well written and experiments are well performed.

Minor points:
1. No scale bar in Fig. 2D and 3A.
2. The title should be “ROCK activity regulates functional tight junction assembly during blastocyst formation in porcine parthenogenetic embryos”. Also please specifically indicate porcine embryo into porcine parthenogenetic embryos in the main text.

Reviewer 2 ·

Basic reporting

No Comments

Experimental design

No Comments

Validity of the findings

No Comments

Additional comments

The authors examined developmental competence and levels of adherens/tight junction constituent proteins after treating porcine four-cell embryos with Y-27632, a specific inhibitor of ROCK. They concluded that ROCK activity may differentially affect assembly of adherens and tight junctions as well as regulate expression of genes encoding junctional proteins.

Overall, the manuscript is well written and technically sound, the results are clearly presented and support the conclusion. It represents a contribution to the field and is highly likely to be cited.

Several specific questions that I have:
1) The abstract seems a bit long, may need to make it concise.
2) Please include embryo number in the related experiments in Figure Legends.
3) Figure 3, change C-D to A-B, also, need to indicate the statistical significance.
4) How did the authors select the different concentrations of ROCK inhibitor used in this study? May need to explain it or cite a reference.

·

Basic reporting

1. The figures can be arranged again. Some images are not necessary. For example, Figure 2B can be deleted. It is similar to Figure 1B. In addition, Figure 2D can be deleted because it has been included in Fig 3A.
2. Line 171: “7.4%” should be changed into “0”. From the Figures and line 33 I did not find 7.4%.
3. Figure 2(D) legend, 100um should be changed into 20um.

Experimental design

The first part in this manuscript overlapped with Zhang's paper(2014). However, this manuscript highlighted that ROCK activity affected assembly of adherens and tight junctions. So I suggest to compress this part. In basic reporting, I mentioned to delete some images and combine some figures.

Validity of the findings

No Comments.

Additional comments

Dr. Kwon et al have attempted to examine developmental competence and levels of adherens/tight junction constituent proteins, as well as expression of their respective mRNAs, after treating porcine four-cell embryos with Y-27632, a specific inhibitor of ROCK. Overall the manuscript is well written and the data are clearly and thoughtfully presented. This is an interesting topic. So I recommend publishing this manuscript after minor revision.

·

Basic reporting

The present study demonstrated that treatment with Y-27643, a specific inhibitor of ROCK, significantly reduced the rate of blastocyst formation, as well as the numbers of total cells in blastocysts. The reason may be that Y-27643 gives negative positive effects on the actin cytoskeleton reorganization and microtubule dynamic through examining the levels of CXADR, OCLN, TJP1, AND CDH1, as well as expression of their respective mRNAs.

Experimental design

The test proves the design is rational and usable.

Validity of the findings

The results indicated that the proposed method is workable and findings are valid.

Additional comments

Fist, if the embryos treated with Y-27643 for 24 h after activation immediately, what are the results like? Whether can affect the in vitro developmental capacity of embryos?
Second, the fourth row images from Figure 3A should be replaced because the quality of blastocyst is much lower than other 20 um treatment groups.

---

## Round 0.2 · accepted · Accept

The comments were addressed.